# Dynamics of cellular states of fibro-adipogenic progenitors during myogenesis and muscular dystrophy

Barbora Malecova [1,8], Sole Gatto [1,9], Usue Etxaniz [1], Magda Passafaro [2,3], Amy Cortez[4], Chiara Nicoletti [1,5], Lorenzo Giordani [1,10], Alessio Torcinaro[6,7], Marco De Bardi[2], Silvio Bicciato[5], Francesca De Santa[6,7], Luca Madaro [2] & Pier Lorenzo Puri [1,2]

Fibro-adipogenic progenitors (FAPs) are currently defined by their anatomical position, expression of non-specific membrane-associated proteins, and ability to adopt multiple lineages in vitro. Gene expression analysis at single-cell level reveals that FAPs undergo dynamic transitions through a spectrum of cell states that can be identified by differential expression levels of Tie2 and Vcam1. Different patterns of Vcam1-negative Tie2$^{high}$ or Tie2$^{low}$ and Tie2$^{low}$/Vcam1-expressing FAPs are detected during neonatal myogenesis, response to acute injury and Duchenne Muscular Dystrophy (DMD). RNA sequencing analysis identified cell state-specific transcriptional profiles that predict functional interactions with satellite and inflammatory cells. In particular, Vcam1-expressing FAPs, which exhibit a pro-fibrotic expression profile, are transiently activated by acute injury in concomitance with the inflammatory response. Aberrant persistence of Vcam1-expressing FAPs is detected in DMD muscles or upon macrophage depletion, and is associated with muscle fibrosis, thereby revealing how disruption of inflammation-regulated FAPs dynamics leads to a pathogenic outcome.

[1] Development, Aging and Regeneration Program at Sanford Burnham Prebys Medical Discovery Institute, La Jolla, CA, 92037, USA. [2] IRCCS Fondazione Santa Lucia, 00142 Rome, Italy. [3] Department of Biology, University of Rome Tor Vergata, 00173 Rome, Italy. [4] Flow Cytometry Core, Sanford Burnham Prebys Medical Discovery Institute, La Jolla, CA, 92037, USA. [5] Department of Life Sciences, University of Modena and Reggio Emilia, Modena, 41125, Italy. [6] Department of Biology and Biotechnology "Charles Darwin", Sapienza University, Rome, 00185, Italy. [7] Institute of Cell Biology and Neurobiology (IBCN), National Research Council of Italy (CNR), 00143 Rome, Italy. [8] Present address: Avidity Biosciences LLC, La Jolla, CA, 92037, USA. [9] Present address: Monoceros Biosystems LLC, San Diego, CA, 92172, USA. [10] Present address: Sorbonne Université, INSERM UMRS974, Association Institut de Myologie, Centre de Recherche en Myologie, 75013 Paris, France. These authors contributed equally: Barbora Malecova, Sole Gatto, Usue Etxaniz. Correspondence and requests for materials should be addressed to B.M. (email: barboramalecova22@gmail.com) or to P.L.P. (email: lpuri@sbpdiscovery.org)

While skeletal muscle stem cells (also referred to as satellite cells—SCs[1]) are unanimously recognized as the direct cellular effectors of muscle regeneration[2,3], other cell types are emerging as critical regulators of SCs[4–8]. These cells include components of the inflammatory infiltrate (e.g., macrophages, eosinophils, and neutrophils)[9,10] and other resident cell types, such as mesenchymal cells endowed with a variable degree of multipotency within the mesoderm-derived lineages[4,11–15]. Among them, muscle interstitial fibro-adipogenic progenitors (FAPs) have been proposed to convert environmental perturbations into cues that coordinate SC activity upon acute injury[16], indicating that these cells provide a highly dynamic functional niche for SCs. Indeed, reciprocal and functional interplay between SC niche components regulates proper execution of essential events during muscle regeneration, such as SC transition from quiescence to activation and eventually differentiation into myofibers. Recent studies have revealed the importance of the timely appearance and clearance of FAPs, in order to restrict their activity within a specific timeframe during the regeneration process[17]. An abnormal persistence of FAPs has been observed in pathological conditions of chronic muscle damage (i.e., muscular dystrophies) associated with persistent inflammation, formation of fibrotic scars, fat deposition, and impaired muscle regeneration[18]. Because of their intrinsic ability to differentiate into fibrotic cells and adipocytes[4,11], FAPs are considered as potential effectors of these maladaptive processes[15]. Moreover, FAPs can also adopt alternative lineages, such as the osteogenic phenotype in response to BMP that appears to mediate muscle heterotopic ossification[19,20]. Overall, FAP's ability to adopt multiple lineages and perform different activities is indicative of their phenotypic and functional heterogeneity in response to environmental signals. Thus, the identification of discrete subpopulations of FAPs and their relative contribution to muscle growth and regeneration in response to physiological or pathological signals is an urgent issue in regenerative medicine.

Here we report the identification of FAP subpopulations, based on Tie2 and Vcam1 expression, that reflect a continuum of cell states in dynamic transition during post-natal myogenesis, muscle repair and disease—the mdx mouse model of Duchenne Muscular Dystrophy (DMD).

## Results

**FAP heterogeneity identified by single cell analysis.** To address the FAP heterogeneity, we have performed gene expression profiling of FAPs at the single cell level using the Fluidigm 96.96 Dynamic Arrays qPCR platform. We compared the profile of FAPs of young (3 months old) wild-type mice, either unperturbed (WT) or at 3 days post notexin-mediated intramuscular injury (WT-inj 3d), the time point at which a substantial increase in FAPs was reported[4,17]. FAPs from 3-month-old dystrophic mice (MDX), the murine model of DMD, provide an experimental setting for chronic muscle injury (Fig. 1a). FAPs were isolated by fluorescence-activated cell sorting (FACS) from hindlimb muscles based on expression of established cell surface markers, as negative for Ter119, CD45, CD31, and α7 integrin and positive for CD34 and Sca-1[4,5,19-21] (Fig. 1a). A total of 87 genes selected for the analysis (Supplementary Table 1) were previously shown to be functionally relevant in FAP biology or have been associated with muscle-derived mesenchymal cells that might phenotypically or functionally overlap with FAPs[4,5,11,13,15–19,22–25].

Principal component analysis (PCA) of the FAP single cell gene expression data revealed a tendency toward clustering among cells from the same experimental condition (Fig. 1b and Supplementary Fig. 1a). Interestingly, PC2 (principal component 2) could distinguish FAPs derived from regenerating muscles

within the context of either acute (WT-inj 3d) or chronic (MDX) injury from FAPs isolated from unperturbed muscles (WT) (Fig. 1b). PC1 appears, instead, to discriminate FAPs from acute and chronic injury (Fig. 1b). We sought to resolve FAP heterogeneity by clustering and plotting the single cell gene expression data using the self-organizing maps (SOM) algorithm. The topological maps (Fig. 1c) are illustrating the average expression level of each gene for each node of the map. The nodes represent clusters of single FAPs with highly similar gene expression profiles, with the number of cells in each node visualized in Supplementary Fig. 1b. The SOM analysis revealed a distinct set of genes correlated within the same cell and showing the highest loadings in PC2 (Fig. 1c, d and Supplementary Fig. 1a), marking putative subpopulations of FAPs. For example, Vcam1 expression shows a pattern similar to Runx2, Lbh, Adam12, Hgf, Lox, Timp1, Pdgfa, Acta2, and Notch3, while high Tek gene expression is associated with cells that express elevated levels of Wnt11, Igfbp5, Myoc, Wnt10b, Smpd3, Bmp6, and Bmp7 genes (Fig. 1c). SOM for all the genes in the analysis are shown in Supplementary Fig. 1c. Co-expression associations among genes within each cell were independently confirmed by a correlation matrix (Fig. 1d). Importantly, the gene expression correlation analysis revealed associations between genes implicated in common signalling pathways within a single cell. Thus, while Vcam1 expression preferentially correlated with a pro-fibrotic gene signature (i.e., Acta2, Pdgfa, Adam12, Lox, Timp1)[15,22,24], high Tek expression preferentially associated with components of the Wnt and Bmp signalling pathways (Wnt5a, Wnt11, Bmp4, and Bmp6) (Fig. 1d). Moreover, we observed that in the SOM analysis, there are nodes containing cells from all experimental conditions, as well as nodes that instead contain only cells from one experimental condition (Supplementary Fig. 1d). Noticeably, the cells clustering in the nodes characterized by a high Vcam1 expression are mostly FAPs isolated from chronic and acute injury environment (MDX and WT-inj 3d), while the cells in the SOM nodes associated with high Tek expression are predominantly WT FAPs isolated from unperturbed muscles (compare Fig. 1c and Supplementary Fig. 1d). We therefore hypothesized that such association of single cell gene expression profiles reflects the existence of discrete FAP subpopulations. Since Vcam1 and Tek (which encodes for the Angiopoietin receptor, Tie2) are cell surface markers, we sought to exploit them for prospective isolation of FAP subpopulations by FACS.

We first simulated the partitioning of FAPs into four predicted subpopulations (subFAPs) defined by the combined levels of Vcam1 and Tek expression on single cell level, using a threshold of 7 log2Ex for both markers (Fig. 1e). The resulting subgroups are Vcam1$^{high}$/Tek$^{low}$ (Vcam1+), Tek$^{high}$/Vcam1$^{low}$ (Tek+), Vcam1$^{high}$/Tek$^{high}$ (double positive, DP), or Vcam1-$^{low}$/Tek$^{low}$ (double negative, DN) (Fig. 1e). By visualizing the experimental conditions on the Tek/Vcam1 gene expression scatterplot (Supplementary Fig. 1e) and by plotting the predicted FAPs subpopulations on the PCA (Fig. 1f), we demonstrate a presumptive association of Tek and Vcam1 expression in FAPs with the different experimental conditions used in this study. Specifically, we observed that FAPs from the two conditions characterized by active muscle regeneration, that is notexin-injured WT mice and MDX mice at early, regenerative stages of disease, were enriched in Vcam1+ subFAPs. Thus, Vcam1 expression appears to mark a putative injury-activated subpopulation associated with regenerating muscles. The Tek+ subpopulation, instead, appears mostly represented in unperturbed muscles of WT mice. The DN putative subpopulation was largely represented in all conditions, while double positive (DP) cells were negligible in our experimental conditions.

**Vcam1 and Tie2 define distinct subFAPs**. We next sought to use FACS to isolate subFAPs, as predicted by the single cell gene expression analysis, by using antibodies against cell surface markers Tie2 (encoded by *Tek* gene) and Vcam1 (Supplementary Fig. 2a–c). FACS profiles of isolated subFAPs (Fig. 2a) revealed dynamic patterns of distribution consistent with our single cell gene expression data (compare Fig. 2b to Fig. 1g). As we isolated subFAPs based on the relative levels of Tie2 protein (encoded by *Tek*) within a continuum of expression profile, and based on the presence or absence of Vcam1, we will refer to Vcam1-negative cells as Tie2$^{high}$ or Tie2$^{low}$ subFAPs. By contrast, Vcam1 expressing subFAPs, which are also expressing low levels of Tie2, will be referred to as Vcam1+ subFAPs. Both Tie2$^{high}$ and Tie2$^{low}$ subFAPs were present, albeit in different proportions, in all conditions tested: unperturbed muscles (WT), regenerating muscles (WT-inj d3) and chronically injured dystrophic muscles (MDX) (Fig. 2a–c). On the other hand, Vcam1+ subFAPs appeared exclusively in the context of injury, either acute (WT-inj d3) or chronic (MDX), and were nearly absent in unperturbed muscles (WT) (Fig. 2a–c). Thus, the FACS profile of FAPs

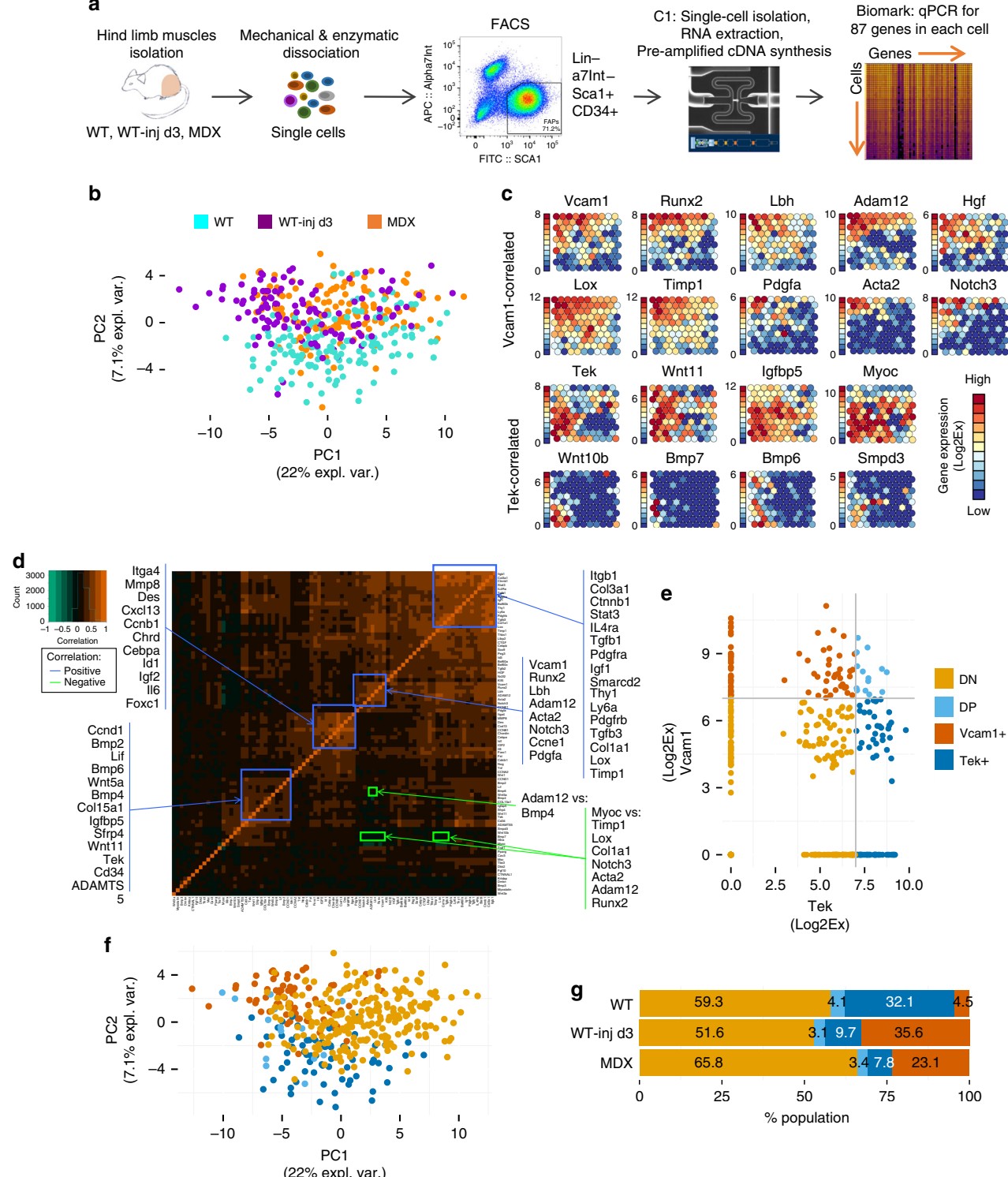

**Fig. 1** Heterogeneous FAPs population consists of distinct subpopulations of cells. **a** Experimental workflow for single cell gene expression analysis. Hindlimb muscles of C57Bl/10 mice were isolated, minced, and enzymatically digested. FAPs were isolated by FACS and loaded on the C1 System (Fluidigm) to extract RNA, reverse transcribe RNA to cDNA and pre-amplify cDNA from each single cell. Real-time qPCR analysis of single cell-derived cDNA was performed on the Biomark platform (Fluidigm) for 87 genes. **b** Principal component analysis (PCA) of single cell gene expression values of FAPs isolated from WT, WT notexin-injured day 3 (WT-inj d3) and dystrophic MDX mice. **c** Self organizing maps (SOM) representation of gene expression in clusters of single FAP cells. Each circle is a cluster of single cells and the fill color represents the level of expression for each gene shown. The expression scale is shown on the left for each gene individually. Expression is measured as $Log_2Ex$ ($Log_2Ex = Ct_{(LOD)}-Ct_{(gene)}$) with LOD = 24 (limit of detection) and Ct = cycle threshold. **d** Correlation matrix of single cell gene expression across all cells. Orange color marks high positive correlation, green color marks high negative correlation. Groups of genes outlined in blue are positively correlated, while the genes outlined in green are negatively correlated. **e** Expression scatterplot of *Tek* and *Vcam1* gene expression. Cutoff is set at 7 $Log_2Ex$ for both genes based on the SOM graph (**c**). Tek and Vcam1 expression levels define the predicted subpopulations, marked as Tek(Tie2)+/Vcam− (Tek+) in dark blue, Vcam1+/Tek(Tie2)− (Vcam1+) in brown, double positive (DP) in light blue and double negative (DN) in gold. **f** The same PCA as in Fig. 1a but with cells color coded based on the FAPs subpopulations predicted in Fig. 1e. **g** Distribution of subpopulations in each experimental condition (n = 2)

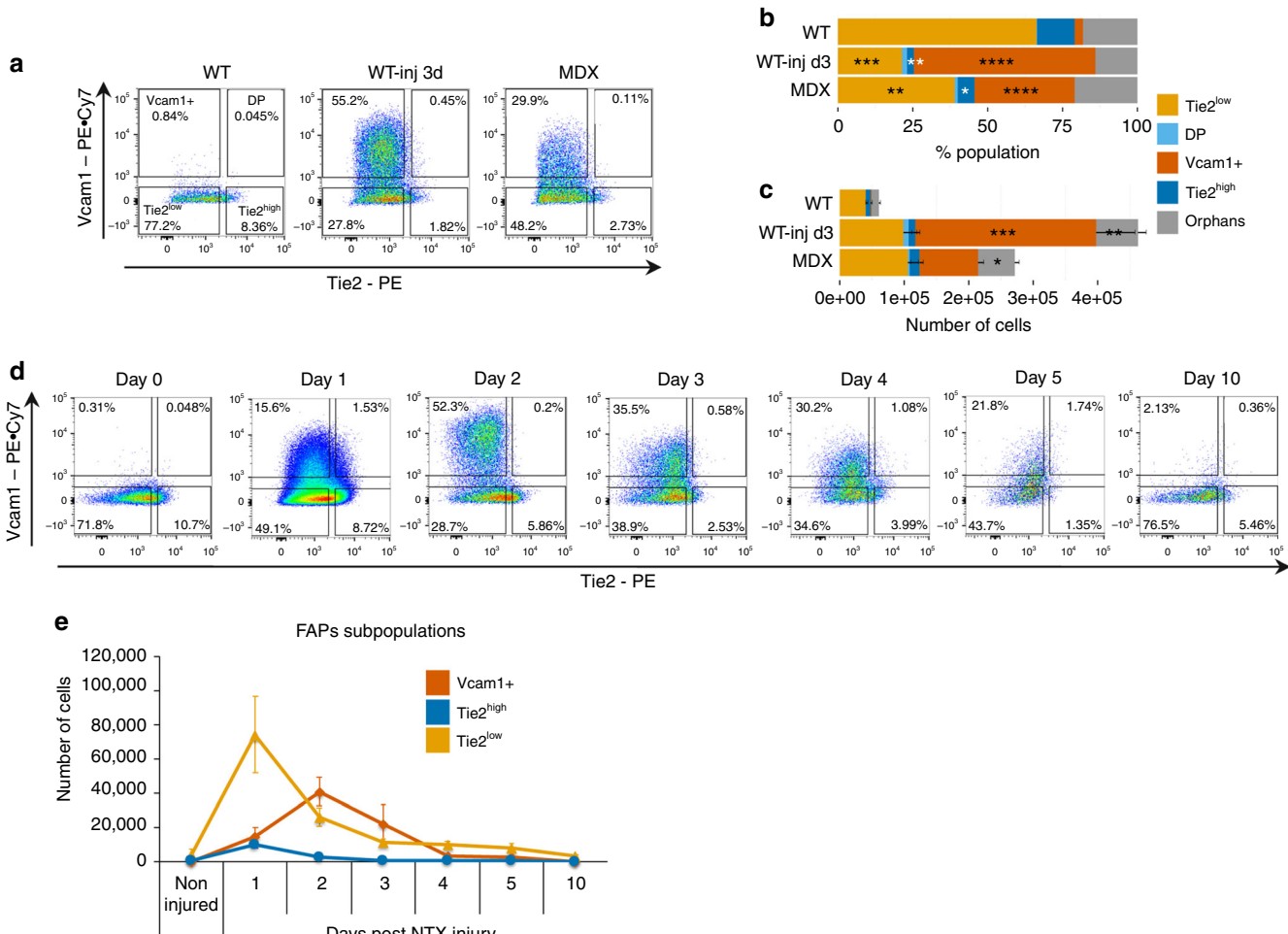

**Fig. 2** Vcam1+ and Tie2-expressing cells are dynamic subpopulations of FAPs. **a** Representative FACS plots of FAPs isolated from hindlimb muscles and analyzed based on Tie2 and Vcam1 expression in wild type (WT), WT notexin-injured day 3 (WT-inj d3) and dystrophic mdx C57Bl/10 mice (MDX). **b** Distribution of subpopulations in each experimental condition by FACS analysis (mean + s.e.m., n = 4, *p-value P < 0.05; **P < 0.01, ***P < 0.001, ****P < 0.0001). Statistical significance was determined by one-way ANOVA with Bonferroni post hoc test, and comparisons to the WT control group are reported. **c** Number of cells in each FAPs subpopulation for each experimental condition by FACS analysis (n = 4, mean + s.e.m., one-way ANOVA, *P < 0.05, **P < 0.01, ***P < 0.001). **d** Representative FACS plots of FAPs based on the expression of cell surface markers Tie2 and Vcam1 following notexin (NTX) injury in WT mice at indicated time points. FAPs were isolated from tibialis anterior (TA) muscles of C57Bl/6J mice. **e** Quantification of cell numbers for the subpopulations of FAPs in TA muscles during the time course in **d** (n = 3 representing independent experiments, mean + s.e.m.)

resembled the pattern of subFAP distribution predicted by single cell gene expression profiling of FAPs (compare Figs. 1g and 2b), with subFAPs formerly named as DN in the single cell gene expression analysis, now being renamed as Tie2$^{low}$, to indicate that they represent cells within a low range of Tie2 expression.

We noted that the proportionally higher abundance of Tie2$^{high}$ cells in unperturbed (WT) muscles, as compared to the other conditions, was only observed in percentage over the whole population (Figs. 1g, 2b), but not in absolute amount (Fig. 2c). In this case, given the great disparity in number of cells in the total

| Table 1 Time course analysis: number of cells per subFAPs during muscle regeneration (related Fig. 2e) | | |
|---|---|---|
|  | Vcam1+ | Tie2high | Tie2low |
| WT | 67 | 547 | 4065 |
| WT-inj 1d | 14,969 | 10,133 | 74,283 |
| WT-inj 2d | 41,009 | 2690 | 25,803 |
| WT-inj 3d | 21,945 | 559 | 11,072 |
| WT-inj 4d | 3301 | 592 | 9768 |
| WT-inj 5d | 2453 | 980 | 7947 |
| WT-inj 10d | 230 | 175 | 3365 |

Average cell number per population is indicated ($n = 3$)

FAPs population between injured and unperturbed muscles, we were not able to accurately estimate the respective proportion of the Tie2high subFAPs among the samples.

qPCR analysis confirmed that Vcam1+ and Tie2high subFAPs isolated from muscles of WT-inj 3d and MDX mice displayed distinct patterns of gene expression that were consistent with those predicted by the single cell analysis. Importantly, the univocal marker of FAPs Pdgfra[11], showed comparable expression levels in all subFAPs (Supplementary Fig. 2d).

To understand the dynamic changes among the identified subFAPs during the course of skeletal muscle regeneration, we have monitored the distribution profile of subFAPs by FACS at different time points following an acute injury by notexin intramuscular injection (days 1, 2, 3, 4, 5 and 10 post injury), and compared it to subFAPs isolated from unperturbed muscles. An immediate, but transient, expansion of Tie2low and Tie2high subFAPs was detected upon acute injury, with a peak observed at day 1 post injury. However, while Tie2high promptly returned to basal levels by day 2, a slower and progressive return to basal levels was observed with Tie2low subFAPs (Fig. 2d, e and Table 1). Interestingly, the emergence of Vcam1+ subFAP was observed at later time points, with a peak between days 2 and 3 post injury and a gradual return to basal levels between days 5 and 10 post injury (Fig. 2d, e and Table 1). Overall, the profiles of subFAPs at day 10 post injury were comparable to those observed in unperturbed muscles (Fig. 2d, e). This evidence indicates that transient and temporally coordinated expansion of distinct subFAPs coincides with muscle repair progression following an acute injury. It also suggests that the clearance of expanded subFAPs upon completion of the regeneration process is an important event for muscle homeostasis. Notably, the observed dynamics of subFAP appearance was not strain specific, as the results obtained with C57BL/10 (Fig. 2a) and C57BL/6J (Fig. 2d) mouse strain could be replicated with mice from ICR/HaJ strain (Supplementary Fig. 2e–h).

**subFAPs exhibit dynamic transcriptional profiles**. To further address subFAP identity and functional relevance, we profiled the gene expression of each subFAP in different experimental conditions by transcriptome analysis. We performed RNA sequencing (RNA-seq) analysis of bulk FAPs and subFAPs (Tie2high, Tie2low, and Vcam1+ only, since the DP subFAPs are extremely rare and therefore not suitable for RNA-seq with the same protocol) by collecting RNA immediately after isolation from hindlimb muscles of WT mice either unperturbed or at 2 time points after acute injury—day 1 and day 3 post injury (WT-inj d1 and WT-inj d3)—as well as from muscles of 3-month-old mdx mice (MDX). RNA-seq gene expression data (normalized counts) showed a high correlation with single cell qPCR gene expression levels (Log2Ex) for all genes marking specific subFAPs (Fig. 1c and Supplementary Fig. 3a). FAPs markers (Pdgfra, Ly6a/Sca-1)

were highly expressed in all subFAPs in both analyses (Supplementary Fig. 3a). Transcriptomic data of subFAPs revealed distinctive transcriptional signatures that discriminated individual subFAPs from each other as well as from the bulk FAPs. The PCA of subFAPs RNA-seq data showed that the samples cluster by discrete subpopulations (Fig. 3a), indicating that the subFAPs identified in this study have distinctive transcriptional profiles regardless the experimental condition they originated from. The gene expression profiles of Tie2low and bulk FAPs tend to partly overlap, consistent with the fact that Tie2low subFAPs account for the majority of the bulk FAPs in most of the conditions.

In order to identify subFAP-specific gene signatures that could assign unique identities to individual subFAPs, we first compared the gene expression profiles of subFAPs vs. bulk FAPs irrespective of the experimental conditions they originated from (Fig. 3b and Supplementary Fig. 3b). We found distinctive transcriptional profiles for each subFAP as well as substantial changes of gene expression within each subFAP, depending on the experimental conditions they were isolated from (Fig. 3b). Differential gene expression (DE) analysis showed that a large number of differentially expressed genes discriminated Tie2high and Vcam1+ from bulk FAPs, while Tie2low subFAPs displayed minimal differences in gene expression, when compared to bulk FAPs (Supplementary Fig. 3b), as it was anticipated by PCA (Fig. 3a). Functional analysis by gene ontology (GO) revealed subFAP-specific gene expression profiles predictive of specialized biological functions (Fig. 3c). For instance, while Vcam1+ subFAPs exhibited global gene expression signatures predictive of high proliferative activity and low adhesion properties, Tie2high subFAPs have profiles predictive of low proliferative activity, but increased propensity to form focal adhesions (Fig. 3d). Tie2low subFAPs displayed specific gene expression profiles indicative of chemotactic activity (Fig. 3d).

We next compared subFAP gene expression profiles across different experimental conditions. To this purpose, we identified differentially expressed genes in each subpopulation and experimental condition compared to the bulk WT (Supplementary Fig. 3c). We then compared the enriched biological functions in all the conditions with Ingenuity Pathway Analysis (IPA) (Fig. 3e–g). This analysis revealed a dynamic regulation of subFAPs during the transition from unperturbed muscles to progressive stages of regeneration following acute injury, or in pathological conditions, such as chronic degeneration/regeneration of mdx muscles (Fig. 3e). Remarkable dynamics was observed during the transition from day 1 to 3 post injury, when both Tie2high and Tie2low subFAPs appeared to acquire transcriptional properties predictive of specific biological functions, such as chemotaxis of blood cells, and of canonical pathways, such as dendritic cell maturation (Fig. 3f, g). This evidence suggests that subFAPs that displayed similar temporal kinetics (Fig. 2d–f) can also share some biological functions presumably related to coordination of inflammation-related events during early regeneration stages. Within this context, we also observed that by day 3 post injury Tie2low, and, to a certain extent, Tie2high subFAPs displayed changes in gene expression that reflect their ability to promote neo-angiogenesis (Fig. 3f). The presence of both Tie2high and Tie2low subFAPs in unperturbed muscles and the large overlap in their gene expression profiles (Supplementary Fig. 3b) suggest that they likely represent a continuum of cellular states in dynamic transition, upon sequential exposure to a plethora of cues generated within the regenerative environment along with the process of muscle repair. Accordingly, during the regeneration process unique transcriptional signatures could discriminate Tie2high and Tie2low subFAPs, such as Tie2high-specific expression of genes implicated in muscle growth, or Tie2low-specific expression of genes implicated in cell spreading (Fig. 3f). Another distinctive

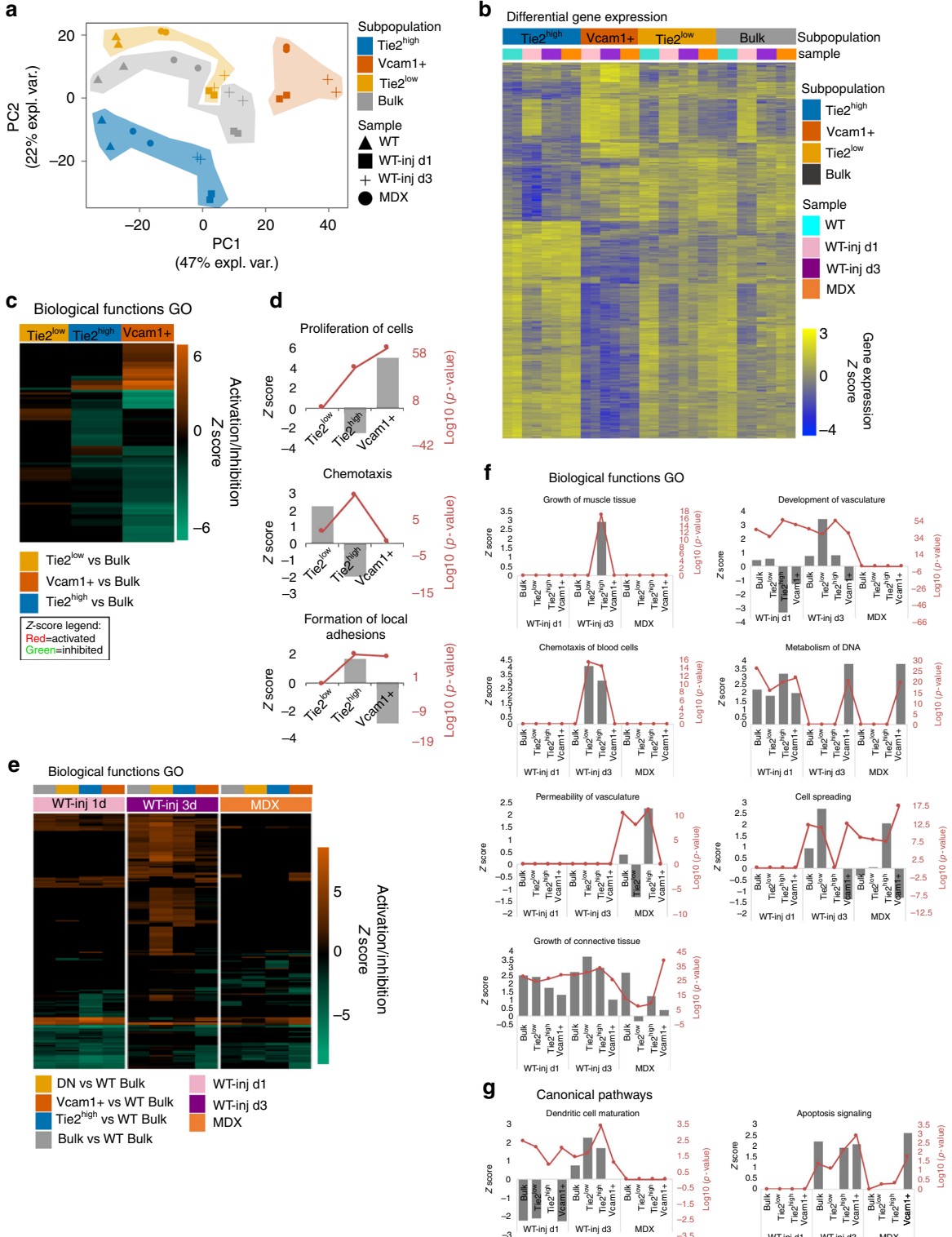

**Fig. 3** SubFAPs have unique transcriptional profiles that are modulated by injury. **a** Principal component analysis (PCA) of gene expression data from RNA-seq analysis of FAPs subpopulations isolated from wild type (WT), WT NTX-injured at day 1 and day 3 (WT-inj d1 and d3) and dystrophic mice C57Bl/10 (MDX). **b** Expression heatmap of genes differentially expressed in FAPs subpopulations (Tie2$^{high}$, Tie2$^{low}$ and Vcam1+) compared to bulk FAPs (adjusted *p*-value <0.001). Gene expression is represented as *z*-score calculated across the rows. **c**, **d** Biological functions predicted to be differentially activated or inhibited in each FAP subpopulation compared to bulk FAPs by IPA comparison analysis. Selected altered functions are shown in **d**. **e**, **f** Biological functions predicted to be differentially activated or inhibited in FAP subpopulations in each treatment condition (WT-inj d1, WT-inj d3 and MDX) compared to bulk WT FAPs by IPA comparison analysis. Selected altered functions are shown in **f**. **g** Selected altered canonical pathways in FAP subpopulations in each treatment condition (WT-inj d1, WT-inj d3 and MDX) compared to bulk WT FAPs by IPA comparison analysis. **d**, **f** and **g** Gray bars represent *z*-score (predicted level of activation/repression of the gene ontology category) with scale on the left. Red lines represent the significance of the prediction in log10 (*p*-value), with scale on the right

biological function between Tie2$^{high}$ and Tie2$^{low}$ subFAPs was the enrichment in the apoptosis signalling observed in Tie2$^{high}$ as well as in Vcam1+ subFAPs at day 3 post injury (Fig. 3g). Cell proliferation (as reflected by an enrichment in the expression of genes implicated in DNA metabolism) was instead the dominant biological process invariably identified in Vcam1+ subFAPs in all the experimental conditions (Fig. 3f).

Overall, the reported changes in the transcriptional profiles of subFAPs along the progression of the regeneration process suggest that individual subFAPs could perform specific tasks within discrete temporal windows, presumably through reciprocal interactions with other cellular components of the regenerative environment. Among these interactive networks, the temporal control of subFAP activity appears of particular interest. Previous works indicated that timely clearance of FAPs by macrophages is required to restrict their activity within a specific window of time during muscle repair in response of acute injury[17]. In this regard, it is interesting to note the opposite pattern of subFAP-mediated control of dendritic cell maturation between day 1 and day 3, which likely reflects a FAPs-dependent regulation of macrophage phenotypic switch that has been implicated in the termination of the regeneration-associated inflammatory activity and the clearance of FAPs[10,26,27]. Interestingly, the subFAP-mediated dendritic cell maturation was not observed in mdx muscles (Fig. 3f, g), consistent with a defective regulation of macrophages previously observed in dystrophic muscles[17]. Likewise, comparison of subFAP gene expression profiles between regeneration post-acute injury and mdx muscles exposed to a chronic damage/regeneration cycles revealed a number of biological processes that were dysregulated in subFAPs isolated from mdx muscles (Fig. 3f, g). For instance, the gene expression profiles that accounted for the ability of subFAPs to promote muscle growth (Tie2$^{high}$ subFAPs), neo-angiogenesis (Tie2$^{low}$ subFAPs), and chemotaxis (both Tie2$^{high}$ and Tie2$^{low}$ subFAPs) were lost in mdx muscles (Fig. 3f). Likewise, a global reduction of the apoptosis signaling was observed in FAPs isolated from mdx mice, as compared to acutely injured WT mice (Fig. 3g).

Unsupervised clustering of RNA-seq profiles separated bulk FAPs and all subFAPs from other muscle-derived cell types, such as SC and macrophages (Supplementary Fig. 3d). Interestingly, tissue-resident fibroblasts, such as cardiac muscle, dermal, lung, and synovial fibroblasts, appeared to cluster separately from subFAPs, suggesting that pro-fibrogenic Vcam1+ subFAPs are distinct from tissue-resident fibroblasts.

Fibrosis is the most deleterious pathological event during the progression of DMD and other chronic disorders[28–32]. As Vcam1+ subFAPs were detected in mdx muscles, and because Vcam1+ subFAP single cell expression profile showed an association with pro-fibrotic genes (Fig. 1c, d), we interrogated the RNA-seq data set for specific enrichment in pro-fibrotic genes in Vcam1+ and other subFAPs. To this purpose, we assembled a list of genes that have been previously implicated in the regulation of fibrosis and myofibroblast identity[22,24,31–35] and determined their relative expression in the subFAPs. An elevated expression of pro-fibrotic genes such as *Acta2, Lox, Adam12, Timp1, Col1a1*, and *Col3a1* was observed in Vcam1+ subFAPs, as also predicted by the single cell gene expression analysis (Fig. 1c and Supplementary Figs. 1c and 4a). Other genes implicated in activation of fibrosis, such as *Snai1, Mmp13, Mmp9, Serpinh1, Stat1*, and *Itgb1* were also enriched in Vcam1+ subFAPs (Supplementary Fig. 4a). The heatmap also illustrated specific dynamics of expression of fibrosis-associated genes in Vcam1+ subFAPs along with the regeneration stages post-injury. For instance, some genes were abundantly expressed at day 1 post injury, but their expression declined at day 3 post injury (i.e., *Plau, Nfkb1, Tgif1, Myc, Cebpb, Serpine1, Smad7, Smad2, Mmp3*,

*Itgb8, Cav1*, and *Eng*). Of note, most of these genes were also expressed at day 1 post injury in Tie2$^{low}$ subFAPs, which was indeed the subpopulation that more closely clustered with Vcam1+ subFAPs (Supplementary Fig. 4a). By contrast, other fibrosis-associated genes were not expressed in Vcam1+ subFAPs at day 1 post injury, but showed a robust expression at day 3 post injury (i.e., *Stat1, Lox, Serpinh1, Acta2, Itgb1, Mmp9, Foxm1*, and *Snai1*). Moreover, few genes were already expressed in Vcam1+ subFAPs, but not in Tie2$^{low}$ subFAPs, and were upregulated in Vcam1+ subFAPs at day 3 post injury (i.e., *Snai1, Mmp13*).

Interestingly, while the large majority of fibrosis-associated genes expressed in Vcam1+ and Tie2$^{low}$ subFAPs at day 1 post injury were not expressed in Vcam1+ subFAPs isolated from mdx muscles, most of the genes upregulated in Vcam1+ subFAPs at day 3 post injury were also expressed in Vcam1+ subFAPs from mdx muscles (Supplementary Fig. 4a). These gene expression dynamics suggests a progressive acquisition of a pro-fibrotic phenotype by Vcam1+ subFAPs during the regeneration process in response to acute injury, with a retention of late-stage (day 3 post injury) "transcriptional signatures" observed in Vcam1+ subFAPs of dystrophic muscles. The intermediate expression levels of some pro-fibrotic genes at day 3 post injury, as well as the detection of few day 1 post injury pro-fibrotic genes in Vcam1+ subFAPs from mdx muscles, are likely accounted for by the presence of asynchronously activated FAPs at different stages of transition through a continuum of cell states, that might reflect the response to repeated cycles of degeneration-regeneration process typically observed in DMD muscles. Moreover, markers of fibrotic deposition in the extracellular matrix (ECM), such as *Col1a1* and *Col1a2*, as well as *Tgfb1* and *Tgfb3*, were preferentially expressed in Vcam1+ (and moderately in Tie2$^{low}$) subFAPs only from mdx muscles. On the other hand, the lack of expression in Vcam1+ subFAPs from mdx muscles of certain genes that were expressed in Vcam1+ subFAPs from regenerating muscles post-acute injury (i.e., *IL13ra2, Timp1*, and *Lox*) suggests that disease-specific transcriptional profiles can discriminate dystrophic Vcam1+ subFAPs from the regeneration-activated counterpart.

Of note, this analysis also revealed clusters of genes mainly involved in inflammation, but only indirectly implicated in fibrosis (*Ccr2, Ccl3, Il1b, Jun, Cxcr4*, and *Tnf*) that were selectively expressed in Tie2$^{low}$ and Tie2$^{high}$ subFAPs at day 3 post injury. Likewise, another cluster of genes preferentially expressed in Tie2$^{high}$ subFAPs was enriched in genes implicated in the BMP signaling, as also anticipated by the single cell analysis (Fig. 1c) and previous works[19,25]. Among these genes, we noticed the expression of a well-known inhibitor of fibrosis, *Bmp7*[36], which might explain the low pro-fibrotic profile of these cells.

Collectively, RNA-seq analysis of subFAPs predicts specific biological functions during regeneration in response to acute injury and reveals pathological gene expression signatures of subFAPs in dystrophic muscles.

**Distinct subFAPs associate with neonatal and adult myogenesis.** Among the functional interactions predicted by the RNA-seq analysis (Fig. 3), the interplay between subFAPs and the inflammatory infiltrate was emerging as a nodal exchange of regulatory signals that determines whether muscles are repaired by SC-mediated regeneration or by fibrotic deposition.

To further investigate the relationship between subFAPs, inflammatory response, SC-mediated myofiber formation, and fibrosis, we compared the subFAPs activation profile during muscle regeneration either in the presence or absence of inflammatory infiltrate. We have monitored subFAPs profiles

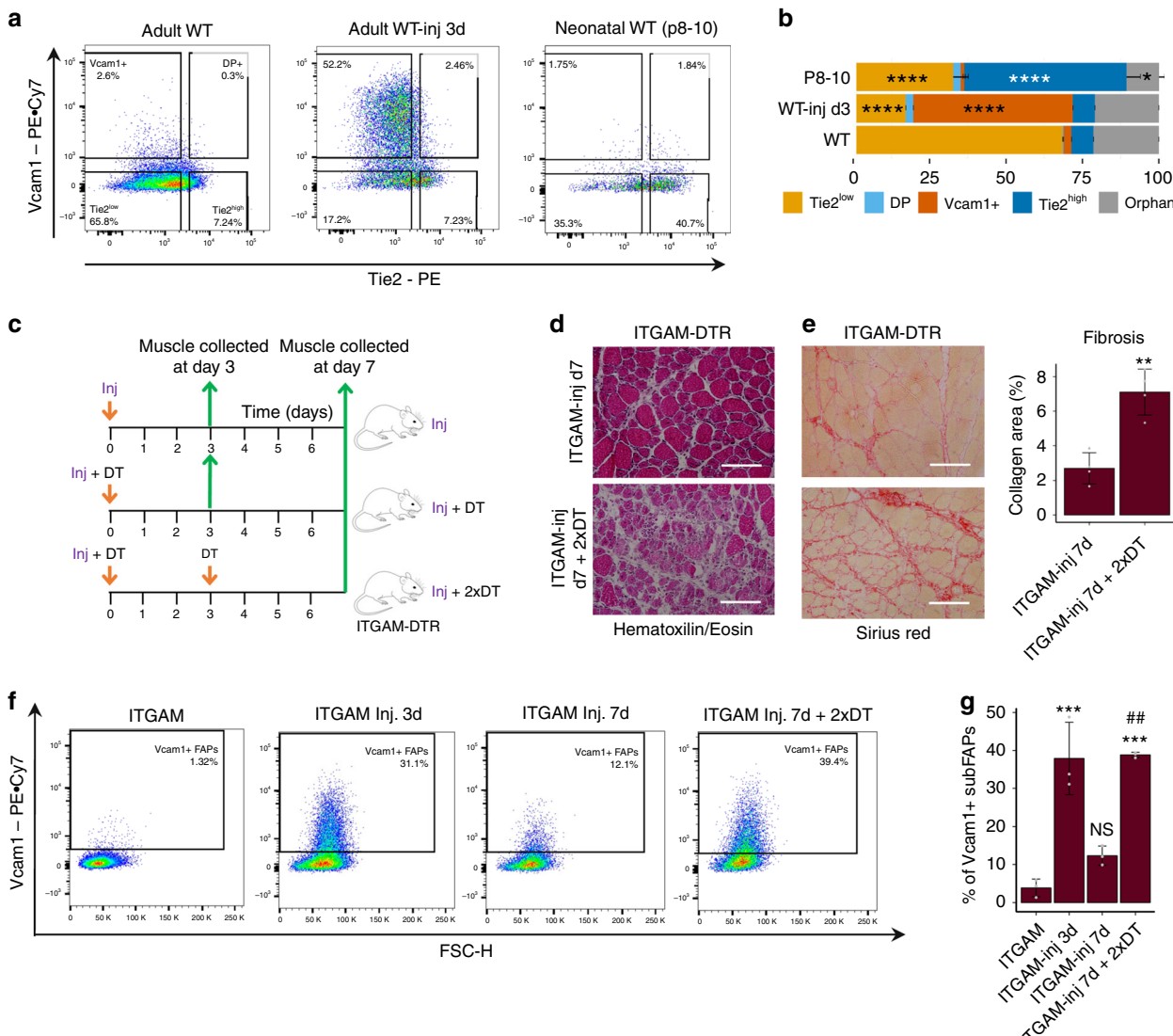

**Fig. 4** SubFAPs are distinctly associated with neonatal and injury-induced myogenesis. **a** Representative FACS plots of FAPs analyzed based on Tie2 and Vcam1 expression in adult wild type (WT), adult WT notexin-injured day 3 (WT-inj d3) and neonatal WT C57Bl/6J mice (postnatal day 8–10). FAPs were isolated from hindlimb muscles. **b** Distribution of subpopulations in each experimental condition by FACS analysis (mean + s.e.m., $n = 6$ (three independent experiments, each including biological duplicates), $*P < 0.05$, $****P < 0.0001$). Statistical significance was determined by one-way ANOVA with Bonferroni post hoc test, and comparisons to the WT control group are reported. **c** Experimental design of the macrophage depletion study. ITGAM-DTR mice were used for this study. Acute muscle injury (Inj) was induced with either NTX (10 μl of 10 μg/ml notexin) or CTX (10 μl of 10 μM cardiotoxin). Macrophage depletion was achieved with diphtheria toxin (DT) injection (12 ng/g) at the time points indicated. Muscles were collected at day 0, 3 or 7 after injury. **d**, **e** Hematoxylin/Eosin (**d**) and Sirius Red (**e**-left panel) stainings of gastrocnemius muscle sections 7 days after an acute injury (ITGAM-inj d7) and 7 days after an acute injury in the context of macrophage depletion (ITGAM-inj d7 + 2xDT). Quantification of fibrosis in muscle sections stained with Sirius Red (**e**-right panel; $n = 4$, mean + s.d., $t$-test, $**P < 0.01$). Scale bar 200 μm. **f** Representative FACS plots profiles of Vcam1+ cells within FAPs population. **g** Percentage of Vcam1+ FAPs in the conditions analyzed in **f**. Statistical significance was determined by one-way ANOVA with Bonferroni post hoc test, and only comparisons to the control ITGAM (*) and to ITGAM-Inj. 7d (#) groups are reported (mean + s.d., $n = 3$ independent experiments, ANOVA, $***P < 0.001$, $##P < 0.01$)

following an acute muscle injury, a condition in which SC activation is associated with the presence of an inflammatory infiltrate, as well as during neonatal muscle growth, in which SC activation is uncoupled from an inflammatory response. We observed a robust activation of Tie2-expressing subFAPs during neonatal myogenesis at days 8–10 after birth, while no activation of Vcam1+ subFAPs was observed, in striking contrast with the Vcam1+ expansion during regeneration post-acute injury (Fig. 4a, b, Supplementary Fig. 4b, see also Fig. 2). This evidence points to inflammatory infiltrate as a major determinant of Vcam1+ subFAP activation.

Previous studies have shown how the inflammatory infiltrate regulates the muscle regeneration process at multiple levels[27,37], including macrophage-mediated clearance of FAPs during the resolution of the regeneration process in order to prevent fibrosis[17]. Defective clearance of FAPs by macrophages has been reported in dystrophic muscles, in association with a chronic inflammation and fibrosis[15]. We used a mouse model of macrophage depletion, the ITGAM-DTR/EGFP (hereafter referred to as ITGAM-DTR) mice, to further investigate the regulatory role of macrophage on Vcam1+ subFAPs clearance. In this mouse model the diphtheria toxin receptor (DTR) and green

fluorescent protein (EGFP) are expressed under the control of the human *ITGAM* (integrin alpha M) promoter (*CD11b*), and macrophage depletion can be induced by intraperitoneal (ip) injection of diphtheria toxin (DT)[27,37]. As the macrophage population tends to be restored by day 4 post-DT administration[27], sequential DT injections every 3 days led to a persistent perturbation of macrophage dynamics, consisting of a reduction of macrophages (Fig. 4c and Supplementary Fig. 4c, d), with a residual population detected that is likely accounted for by resumption of early pro-inflammatory (M1) macrophages[25]. This led to an altered clearance of cell infiltration, persistence of necrosis and formation of fibrotic lesions that were detected in muscles at 7 days post injury (Fig. 4d, e). The impact of macrophage depletion on Vcam1+ subFAPs clearance from muscles following acute injury was then evaluated by FACS-mediated cell isolation from muscles of ITGAM-DTR mice exposed to injury followed by sequential injections of DT or control (PBS). DT-mediated alteration of macrophage dynamics resulted in the aberrant retention of Vcam1+ subFAPs in muscles at day 7 post injury (Fig. 4f, g and Supplementary Fig. 4e). This evidence further emphasizes the functional relationship between the inflammatory infiltrate, the regulation of Vcam1+ subFAP and the formation of fibrotic scars during chronic, pathological regeneration (i.e., DMD).

Finally, we evaluated whether the dynamics of subFAPs typically observed in unperturbed conditions was skewed toward an enrichment in Vcam1+ subFAPs in diaphragm of mdx mice at early stages of disease progression (6 weeks old mice), compared to age-matched control (WT) mice. The diaphragm is an involuntary muscle subjected to continuous contraction cycles that permit breathing. As such, it is invariably used from birth and is therefore the earliest muscle to develop fibrosis during DMD progression, showing the most pronounced accumulation of fibrotic areas[15]. Diaphragms of 6-week-old-WT mice exhibited a higher ratio between Tie2[high] and Tie2[low] subFAPs, compared to the limb muscles used for the previous analysis, presumably because of more intense contractile activity and younger age (compare Supplementary Fig. 4f-g with Fig. 2d). Remarkably, mdx diaphragms exhibited a significant reduction in percentage of Tie2[high] subFAPs and a proportional increase in Vcam1+ subFAPs (Supplementary Fig. 4f, g).

## Discussion

Single cell gene expression profiling of FAPs revealed that the relative expression levels of two cell surface markers, Tie2 and Vcam1, can be exploited for the prospective isolation by FACS of three subpopulations of FAPs that exhibit distinct dynamics of appearance in muscles upon specific environmental cues (neonatal muscle growth, adult homeostasis, and regeneration by acute or chronic injury).

While Tie2-expressing subFAPs (either Tie2[low] or Tie2[high]) account for the vast majority of resident FAPs in unperturbed muscles, Vcam1 expression marks an injury-activated subpopulation that was functionally associated with the presence of an inflammatory infiltrate. We found that the relative levels of Tie2 expression resolve Tie2-expressing FAPs into two subpopulations, Tie2[low] and Tie2[high] subFAPs that show distinct dynamic profiles during muscle regeneration following acute injury or during neonatal muscle growth. The overlap in the gene expression profiles of all subFAPs (including common FAP surface markers, such as Sca1, Pdgfrα, and CD90)[4,11,21,38] together with the emergence of transcriptional signatures that discriminate the various subFAPs upon muscle perturbation suggest that subFAPs range through a spectrum of cell states that are in dynamic transition. SubFAPs are regulated by signals that also

trigger SC expansion and differentiation into myofibers, either within the neonatal muscle growth or during adult life (i.e., muscle repair in response to acute injury). This is consistent with previous studies from Goldhamer lab that identified Tie2+ progenitor cells also expressing the FAP markers Pdgfrα and Sca-1 and displaying an adipogenic potential[19,25]. It is therefore likely that during development the bulk of FAPs consists of Tie2-expressing cells, and an initial bifurcation from this lineage could be provided by an expansion of Tie2[high] subFAPs observed in concomitance with SC amplification and muscle growth during neonatal life. While the upregulation of Tie2 (coding for the Angiopoietin receptor) suggests that enhanced response to Angiopoietin might contribute to this process, future studies are required to determine the molecular regulation of Tie2[low]-to-Tie2[high] transition.

The appearance of Vcam1+ subFAPs in adult life coincides with muscle perturbations associated with an inflammatory response, e.g., regeneration of injured myofibers. Thus, two distinct types of FAP activation can occur during SC expansion and formation of new muscles, in response to different stimuli; while neonatal myogenesis appears dominated by a selective activation of Tie2[high] subFAPs, during adult life injury-activated muscle regeneration triggers a more complex response that includes sequential and partially overlapping waves of macrophages, as well as other inflammatory cells, and a spectrum of FAP functional states that include both Tie2[high] and Vcam1+ subFAPs.

The different abundance and kinetic of appearance of Tie2[high] and Vcam1+ subFAPs, together with the dynamic gene expression profiles revealed by RNA-seq analysis of subFAPs, suggest that they represent distinct cellular states of a functional continuum within a broader cell population (bulk FAPs) that enables muscles to adapt to the diversity of demands imposed by various types of muscle perturbations. In particular, the emergence of Vcam1+ subFAPs during adult life, in association with the inflammatory response to muscle injury, indicates a transition toward a pro-fibrotic state consistent with the requirement of a transient deposition of extracellular matrix to promote asymmetric division of SC within a defined temporal window, a mechanism that enables adult skeletal muscles to regenerate in response to repeated injuries[39,40]. Of note, this process is compromised in SC from DMD muscles, because of deregulation of intrinsic and/or extrinsic properties[6,41]. We propose that Vcam1+ subFAPs accumulation in DMD muscles and the ensuing formation of fibrotic scars, possibly caused by pathological behavior of macrophages, can also contribute to impair SC activity at late stages of disease.

The association of Vcam1+ subFAPs with inflammation and fibrosis, and their pro-fibrotic gene expression profile, suggest that they are candidate cells for the origin of myofibroblasts—the direct effectors of fibrosis[42]. However, we note that Vcam1+ subFAP can be discriminated from resident fibroblasts by their ability to undergo adipogenesis in vitro, upon culture with adipogenic medium, a unique biological property that currently defines FAPs. Our experience with culturing of subFAPs ex vivo isolated from their physiological context indicates that subFAPs rapidly lose their identity and are extremely unstable. While the instability of subFAPs in vitro is consistent with their dynamic transition from one state to another in response to regulatory signals, it should also warn against potential biases in the interpretation of data generated from prolonged ex vivo cultures of FAPs, where signals from serum growth factors replace the regulatory cues in vivo. While these caveats have complicated so far the identification of specialized subpopulations, our analysis of the transcriptional profile of FAPs immediately after their isolation have revealed the existence of subFAPs that represent a spectrum of cellular states highly responsive to environmental cues, such as inflammation. Indeed, our data sets from the single

cell gene expression analysis and RNA-seq of subFAPs provide criteria for a molecular definition of subFAPs in dynamic transition that would not be otherwise captured by current standard assays, such as adipogenic differentiation.

This study discloses a previously unappreciated complexity of FAP biology by revealing their dynamic specialization into sub-FAPs in physiological and disease conditions, and provides the foundation for future strategies targeting specific subFAP in order to further determine their role in vivo.

## Methods

**Animals and in vivo procedures**. All protocols were approved by the Sanford Burnham Prebys Medical Discovery Institute (SBP) Animal Care and Use Committee (IACUC) and the Italian Ministry of Health, National Institute of Health (IIS) and Santa Lucia Foundation (Rome). Mice were age, sex, and strain matched. Normal wild-type (wt) C57Bl/10 and mdx (C57Bl/10 and C57Bl/6J) mice were purchased from the Jackson laboratory. C57Bl/6J and ICR/HaJ mice colonies were maintained in SBP vivarium. Acute injury and muscle regeneration were induced in 2–3-month-old wt C57/BL10 or C57/BL6J by intramuscular injection of 10 μl of 10 μg/ml notexin (NTX, Sigma) into the tibialis anterior (TA), gastrocnemius or quadriceps. Alternative acute muscle injury was performed by intramuscular injection of 10 μl of 10 μM cardiotoxin (CTX, Sigma) into TA muscles. Mice were killed post injury at the time points indicated in figures. For postnatal myogenesis studies, hindlimb muscles from 8 to 10 days old pups were utilized. For experiments using diaphragms, 6-week-old normal wt C57Bl/6J and mdx C57Bl/6J mice were used.

ITGAM-DTR mice were obtained from the Jackson Laboratory. Tibialis anterior (TA) and gastrocnemius (GA) muscles of young (8–12 weeks) ITGAM-DTR mice were injured by intramuscular (IM) injection of notexin (NTX) or cardiotoxin (CTX), as described above. The administration of the diphtheria toxin (DT) (Sigma) was done by intraperitoneal (ip) injection of 200 μl of DT solution (12 ng/per gram of body weight; diluted in PBS) prior to muscle injury (3–4 h before NTX or CTX administration). As monocyte/macrophage population is restored by day 4 following a single intraperitoneal dose of DT, mice were subjected to a second injection of DT at day 3 after injury. DT- and vehicle-treated mice were killed at 7 days post injury and hindlimb muscles were collected for cell isolation and histological assessment of muscle regeneration and fibrosis progression.

**FACS-mediated isolation of FAPs and Macrophages**. FAPs for single cell analysis were isolated from diaphragm and hindlimb skeletal muscles (as indicated on the figures) by fluorescence-activated cell sorting (FACS), as previously described[20]. Briefly, hindlimb muscles were mechanically minced and enzymatically digested in FACS buffer (HBSS (Gibco) containing 0.4 mM CaCl$_2$ and 5 mM MgCl$_2$, 0.2% (w/v) bovine serum albumin (BSA) (IgG-free, protease-free, Jackson ImmunoResearch)) containing 2 mg/ml (0.45 U/ml) Collagenase A (Roche) and 2.4 U/ml Dispase I (Roche) for 45–50 min at 37 °C in a rotating water bath. Cell suspension was diluted with FACS buffer and filtered through 70 μm nylon cell strainer (BD Falcon). Cells were immune-labeled with primary antibodies anti-CD31-PacificBlue (RM5228, Life Technologies, 4 μg/ml), anti-CD45-eFluor450 (clone 30-F11, eBioscience, 4 μg/ml), anti-Ter119-eFluor450 (clone TER-119, eBioscience, 4v μg/ml), anti-Sca-1-FITC (clone E13-161.7, BD Pharmingen, 10 μg/ml), CD34-Alexa Fluor 647 (clone RAM34, BD Pharmingen, 10 μg/ml) and anti-α7 integrin-PE (clone R2F2, AbLab, 2 μg/ml) for 30 min at 4 °C, in a FACS buffer. Cells were washed and resuspended in FACS Buffer and filtered through 40 μm strain filter. FxCycle™ Violet Stain (Life Technologies) was used to label dead cells. Flow cytometry analysis and cell sorting were performed on a FACSAria instrument, and the data were analysed by FACSDiva 6.1.3 software and by FlowJo. Live FAPs were isolated as Ter119-/CD45-/CD31-/α7-integrin-/ CD34+/Sca-1+ cells.

FAPs subpopulations were isolated from diaphragm and hindlimb muscles as described above and sorted by FACS after immune labeling as follows, using the primary antibodies anti-CD31-PacificBlue (RM5228, Life Technologies, 2 μg/ml), anti-CD45-eFluor450 (clone 30-F11, eBioscience, 4 μg/ml), anti-Ter119-eFluor450 (clone TER-119, eBioscience, 4 μg/ml), anti-Sca-1-FITC (clone E13-161.7, BD Pharmingen, 10 μg/ml), anti-α7 integrin- Alexa Fluor 647 (clone R2F2, AbLab, 4 μg/ml), anti-Tie2-PE (clone TEK4, BioLegend, 8 μg/ml), and anti-Vcam1-PE•Cy7 (clone 429 MVCAM.A, BioLegend, 1 μg/ml) for 45 min at 4 °C, in a FACS Buffer containing 10% goat serum. After the immune-staining cells were processed for FACS analysis as described above. FMO controls were prepared with aliquots of cells. Single color controls were prepared using UltraComp eBeads (eBioscience).

Flow Cytometry studies were performed at the Sanford Burnham Prebys Medical Discovery Institute Flow Cytometry Core using LSRFortessa X-20 cell analyzer and FACSAria sorter (BD), and the data were analysed by FACSDiva 6.1.3 software.

For macrophage isolation, hindlimb skeletal muscles were processed as described above and analyzed by FACS in LSRFortessa X-20, but also in DAKO-Cytomation MoFlo High Speed Sorter, analysing data by Summit V4.3 software (Beckman Coulter). The staining- protocol is identical and the following antibodies were used: anti-CD45-eFluor450 (clone 30-F11, eBioscience), anti-CD11b-PE-Cy7

(clone M1/70, eBioscience, 1:200), anti-F4/80-PE (clone BM8, eBioscience, 1:50), and anti-GR1-e780 (clone RB6-8C5, eBioscience, 1:200).

All FACS data were further analysed by FlowJo version 10.0.4 (FlowJo LLC).

**Single cell capture and cDNA preparation**. C1 Single-Cell Auto Prep System (Fluidigm) was used for isolation of FAPs single cell RNA (isolated from male mice), following the manufacturer's protocol (#100-4904, Fluidigm), using 10–17 μm medium size C1 IFC. Integrity of captured FAPs on the C1 IFC was determined under the confocal microscope. Captured cells were lysed and the RNA was reverse transcription and cDNA was pre-amplified using DELTAgene Assays targeting 87 selected genes (Supplementary Table 1).

**BioMark real-time PCR on single cell cDNA**. Collected pre-amplified cDNA was analysed by real-time PCR using Fast Gene Expression Analysis protocol with EvaGreen® on the BioMark HD System with 87 selected gene-specific assays (Supplementary Table 1) (protocol #68000088, Fluidigm), skipping the exonuclease I treatment of C1 collected cDNA[20].

**Single cell gene expression data analysis**. Data analysis after BioMark qPCR is described in detail in our Methods and Protocols chapter[20]. Briefly, we manually checked for data quality in the BioMark RT-PCR analysis software and we extracted the data. Data manipulation was performed in R, where we first eliminated all the data marked as "Fail" in the CtCall.

During the quality control of single cell gene expression data analysis, genes that failed qPCR reactions and cells that either did not express FAP identification markers (Sca1 and Pdgfra) or expressed SC markers (Itga7) or endothelial marker (Pecam1/CD31) were excluded from the data set in order to avoid the inclusion of cell types different from FAPs in the analysis. We then calculated the gene expression level as Log$_2$Ex (Log$_2$Ex = Ct$_{(LOD)}$-Ct$_{(gene)}$) and set low expressing cells (Log$_2$Ex < 0) to zero. We used a LOD of 24, as estimated by Livak et al.[43]. All graphs were produced in R. SOM analysis was performed with kohonen package in R (v2.0.19).

**RNA isolation and qPCR analysis**. RNA from FAPs was extracted using miR-Neasy Micro kit (Qiagen) following the manufacturer's protocol, including DNase treatment of the samples. cDNA was synthetized using QuantiTect Reverse Transcription kit (Qiagen) and analysed by real-time quantitative PCR (RT-PCR) using SYBR Green PCR Master Mix (Applied Biosystems), using primers provided in Supplementary Table 2. Relative expression was calculated using $2^{-\Delta ct}$ method[44], Gapdh housekeeping gene expression was used to normalize gene expression.

**RNA sequencing**. SubFAPs and FAPs were isolated from hindlimb muscles of C57Bl/10 male mice in two independent experiments. The number of cells ranged between 7700 and 230,000. RNA from FAPs was extracted using miRNeasy Micro kit (Qiagen) following the manufacturer's protocol. RNA was shipped to the sequencing facility as dry pellet in RNAstable® (Biomatrica). The libraries for sequencing were prepared using Ovation SoLo RNA-Seq System by NuGEN.

For each experimental condition two independent experiments were carried out for the isolation of RNA. All biological duplicates are from different cohorts of mice, sorted at different times.

**RNA sequencing data processing**. For sequencing alignment, we used the human reference genome assembly GRCm38/mm10 (http://ftp.ensembl.org/pub/release-76/fasta/mus_musculus/dna/), while for transcriptome annotation we used the version 85 of the GRCm38 (http://ftp.ensembl.org/pub/release-85/gtf/mus_musculus/Mus_musculus.GRCm38.85.gtf.gz).

We used the FASTQC package (v0.11.3) to assess the quality of sequenced libraries. All passed quality control.

Because, we were using the Ovation SoLo RNA-Seq System by NuGEN, we adhered to the manual suggestions and trimmed 5 bases from the 5′ of the sequences.

Reads were mapped to the reference genome using TopHat2 v.2.1.1[45]. We used the following non-default TopHat2 parameters: -p 48 -g 1 --library-type fr-firststrand. The number of mapped reads ranged between 23 and 44 × 10$^6$ and the percentage of mapping was between 79 and 92%.

The quality control of the reads distribution along transcripts was performed using infer-experiment.py from RSeQC package v2.6.3[46]. All samples had a uniform distribution of reads along transcripts.

The sequenced read counts per annotated gene were derived with the use of htseq-count script distributed with HTSeq v0.5.4p5[47]. We used the R library package DESeq2 v.1.12.4[48] for measuring differential gene expression between two different cell conditions, considering the two RNA-Seq experiments as biological replicates. We picked genes with adjusted p-value < 0.01.

Gene ontology analysis was performed using Ingenuity pathway analysis (IPA; http://www.1ingenuity.com). All graphs were produced in R.

Public RNA-seq data from mouse tissue-resident fibroblasts were downloaded from GEO with accession numbers GSM1223640 and GSM1223641 (cardiac

fibroblasts), GSM521651 (lung fibroblasts), GSM2500874, GSM2500875 and GSM2500876 (synovial fibroblasts), and GSM2067698, GSM2067699 and GSM2067700 (skin fibroblasts) (see Supplementary Table 3). Data sets from Satellite cells (SRR7075694, SRR7075695, SRR7075698, SRR7075699, SRR7075710, SRR7075711), and macrophages (SRR7075704, SRR7075705, SRR7075706, SRR7075707, SRR7075708, SRR7075709) were kindly provided by S. Consalvi and V. Saccone (Iannotti et al., in preparation). Data were aligned on mm10 version of UCSC mouse genome with Tophat2 and quantified with htseq-count. Counts data from all conditions were filtered based on their raw count, keeping only those where the sum of the counts for all samples is higher than 1, then normalized and logged with DESeq2 v.1.4.5 rlog function, and samples were clustered according to the 50% most variable genes with hclust function, using Pearson correlation coefficient as distance and complete as linkage, respectively, in R-3.1.0.

**Histological and immunofluorescence analyses.** Muscle cryosections were fixed in 4% paraformaldehyde (Sigma-Aldrich, St. Louis, MO, USA) and stained with Haematoxylin/Eosin solution (Sigma-Aldrich), according to standard procedure.

Fibrosis was measured using Sirius Red staining protocol. Briefly, muscle cryosections were fixed in bouin solution 1 h at 56 °C. Sections were stained in Picro-Sirius red solution for 1 h at RT protected from light. After a brief washing in acidified water, sections were fixed in 100% ethanol and the final dehydration was performed in xylene 100%. Sections were mounted with EUKITT® and visualized using a Nikon Eclipse 90i. Fibrotic index was calculated as the percentage of red positive areas using Image J software. The figures reported are representative of all the examined fields.

**Statistics.** Statistical analysis was performed in GraphPad Prism 7 (GraphPad Software Inc, La Jolla, CA, USA). Comparisons between two groups were tested using an unpaired $t$-test. Comparisons between multiple groups were tested using one-way ANOVA and Bonferroni post hoc tests. Differences were considered significant at the $P < 0.05$ level.

## Data availability

RNA-seq data supporting the findings of this study have been deposited in the GEO data repository under the accession code GSE100474. Other data that support the findings of this study are available under the accession codes GSM1223640, GSM1223641, GSM521651, GSM2500874, GSM2500875, GSM2500876, GSM2067698, GSM2067699, GSM2067700, SRR7075694, SRR7075695, SRR7075698, SRR7075699, SRR7075704, SRR7075705, SRR7075706, SRR7075707, SRR7075708, SRR7075709, SRR7075710, SRR7075711.

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

## Acknowledgements
This work was supported by NIH P30 pilot grant P30AR061303 and CIRM training grant TG2-01162 to B.M.; NIH grants R01AR056712, R01AR052779, P30AR061303, MDA, AFM and EPIGEN grant to P.L.P.; C.N. was the recipient of an EPIGEN travel fellowship for training of young scientists in foreign laboratories. We thank Emanuela Aleo at the Institute of Applied Genomics in Udine, Italy, for the RNA-seq library preparation and sequencing; Sonal Naik at Sanford Consortium for Regenerative Medicine in La Jolla, CA, for the assistance with the Fluidigm microfluidic platform for single cell gene expression experiments; Yoav Altman for flow cytometry related discussions and advice; Simon Melov and James Flynn for preliminary analysis; Buddy Charbono, Tomi Omel, Elizabeth Chappell, Diana Sandoval and Andy Vasquez from the Animal Facility at the SBP, and Alessandra Dall'Agnese and Tom Roberts for critical discussion.

## Author contributions
Project conception and design B.M. and P.L.P; Single cell gene expression profiling B.M.; Bioinformatics Analysis S.G. C.N. and S.B.; Experimental contribution B.M., S.G., U.E., L. G., L.M., M.P. and A.T.; Flow Cytometry A.C and M.D.B.; Manuscript writing P.L.P., B.M., S.G. and U.E.; Resources P.L.P. and F.D.S.; Funding P.L.P. and B.M.

## Additional information

**Competing interests:** The authors declare no competing interests.

