## [Peer Review File · Nature Communications]

Reviewers' Comments:

Reviewer #1:

Remarks to the Author:

The manuscript has been revised substantially in response to the first round of review. I feel that in its present form, if data are replicable, it represents an important and interesting study that makes a valuable contribution to the field.

The only issue that was not completely addressed relates to rigor and reproducibility. My previous review noted that sample sizes needed to be increased in certain studies (involving experimental manipulations), and that it was important that studies be replicated to ensure reproducibility. In one case, the authors replicated the observation of VCAM1 induction in response to injury in another strain of mouse, adding to confidence in this observation. This is good. However, there are 2 experiments for which replication still needs to be confirmed, and in the latter sample size needs to be increased:

1. The time course study (Fig. 3 E and F). This appears to have been done only once, with sample size of 3. Has it been repeated, and did repeats provide similar results?

2. The macrophage depletion experiment (Fig. 4 C-H). Have these studies been replicated, and how many times? In addition, while panels D and E are based on n=4 data (an undesirably low sample size, but something often seen in studies of this nature) panels F and G are based on n=2. This is not an acceptable sample size.

Please note that by replication I mean something distinct from sample size; I am referring to replication of an entire study, with its own new groups.

Also, I was not able to find the figure legends to the supplemental figures. Therefore, I was unable to evaluate what sample sizes were used in these experiments, nor whether experiments had been replicated.

Sample size issues may apply to some of this data as well, for example S4 C,D.

The Venn diagram figure, appearing after Fig. 1 at the end of the pdf was labeled Suppl. Figure 8. However, it appears to be out of place or mislabeled, as there are no supplemental figures 5-7.

Reviewer #3:

Remarks to the Author:

This is the revised version of Nat. Cell Biol. One of major issues raised by this reviewer was the differentiation ability of each subFAPs. In this revised version, authors sincerely addressed to this issue. Authors also provided new data in SFig.3d (clustering of RNA-seq profiles). Based on these results, authors mentioned that each subFAPs is discriminated them from tissue fibroblasts. However, the frequency of adipogenic-induction was too low (only 20-30%). In addition, molecular signature of fibroblast might be strongly affected by their environment. In this case, cardiac fibroblast data alone might be not enough. More data derived from other tissues (non-muscle tissues) will strengthen authors' finding. Reviewer strongly recommends to include the data indicating the results of adipogenic differentiation ability of subFAPs in the main or supplemental Figure.

In dystrophic muscle, there are many fibroblasts and myofibroblasts. In this study, authors used limb muscle of mdx mice which show less dystrophic change. Thus mdx sample might not include fibroblasts and myofibroblasts. Does subFAPs have the possibility to include fibroblasts and myofibroblasts? In order to confirm this question, diaphragm sample of mdx might provide

important information.

As reviewer pointed out, the difference between acute and chronic change of FAPs is important when considering the onset or progression of pathological changes of skeletal muscle. Authors described signatures of each subFAPs. For example, Vcam1+ and Tie2low exhibit a pro-fibrotic gene expression profile. Fibrosis-related genes are highly enriched in Tie2-low and Tie2-high, but they also express anti-fibrotic gene, BMP7. Authors explained that BMP7 might contribute to non-fibroblastic role of Tie2-low and high FAPs. However, there is no data indicating the direct evidence for the correlation between the genes and pathological change including fibrosis. Without these data, this work seems to be descriptive.

Please see additional comments.

Specific comments

1: Related to FACS analyses

Authors RE: We are aware of this discrepancy. However, given that we have generated a statistically robust dataset (biological triplicates), the only explanation we can provide at this stage regards most likely the different intensity/extent of an acute injury. As we are comparing our data with only one report, and we believe this contention should remain open to further experimentation from other investigators.

Authors have to stain tissue samples with anti-Pdgfra antibody, a marker for all subFAPs. Authors might observe many Pdgfra+ cells in inj-3d than these in inj-1d. The result will let authors know that the previous report is not in the case of this study or authors' FACS analyses did not perform correctly.

2. FACS profiles of WT inj3d are different among in Figure 2a, 2d, and 4a. Are there any experimental differences among these samples?

3. In GO 'metabolism of DNA', Vcam1+ subFAPs shows high Z-score. However, the result of WT-inj d3 Bulk shows no score. Regardless of the fact that 60-70% 'WT-inj d3 Bulk' consist of Vcam1+ subFAPs', why did WT-inj d3 Bulk show no score?

4. Suppl. Figure 8 and Suppl. Figure 3c is same results. Please remove the S. Fig.8

Introduction to the rebuttal letter

We thank the reviewers for their insightful comments and feedback on our manuscript entitled "A spectrum of cellular states within Fibro-Adipogenic Progenitors (FAPs) upon physiological and pathological perturbations of skeletal muscle".

This manuscript was originally submitted to Nature Cell Biology, and after the first cycle of revision it was transferred to Nature Communications.

Reviewers 1 and 3 from the first submission to Nature Cell Biology have seen the rebuttal to their comments on the original submission and have raised few additional points that we have addressed as described below.

Reviewer #1 (Remarks to the Author):

The manuscript has been revised substantially in response to the first round of review. I feel that in its present form, if data are replicable, it represents an important and interesting study that makes a valuable contribution to the field.

The only issue that was not completely addressed relates to rigor and reproducibility. My previous review noted that sample sizes needed to be increased in certain studies (involving experimental manipulations), and that it was important that studies be replicated to ensure reproducibility. In one case, the authors replicated the observation of VCAM1 induction in response to injury in another strain of mouse, adding to confidence in this observation. This is good. However, there are 2 experiments for which replication still needs to be confirmed, and in the latter sample size needs to be increased:

1. The time course study (Fig. 3 E and F). This appears to have been done only once, with sample size of 3. Has it been repeated, and did repeats provide similar results?

Re: The time course experiment (the subFAPs profiles during early stages of regeneration followed by an acute notexin-mediated muscle injury) was performed three times on different days with an independent cohort of mice. The data on figure 2e,f represent the mean of these three independent replicates. The individual FACS profiles from each independent experiment are available for reviewers upon request.

2. The macrophage depletion experiment (Fig. 4 C-H). Have these studies been replicated, and how many times? In addition, while panels D and E are based on n=4 data (an undesirably low sample size, but something often seen in studies of this nature) panels F and G are based on n=2. This is not an acceptable sample size. Please note that by replication I mean something distinct from sample size; I am referring to replication of an entire study, with its own new groups.

Re: We would like to apologize that we could not increase the sample size (Fig. 4f-h) by the time of resubmission, as we decided to use all the available animals to validate the macrophage depletion (Suppl. Fig. 4c-d), which was also requested by reviewers. We have performed the experiment to reach the sample size n=3, as requested by the reviewer. We also note that we were able to replicate the experiments in two different labs with different personnel/instrumentation, which definitely increase the confidence in the reproducibility of these data (see below).

Three independent FACS experiments of Vcam1+ subFAPs analysis during regeneration after acute injury in macrophage-depleted mice (ITGAM-DTR mice +/- DT treatment)

Figure legend:

The individual FACS profiles from each independent experiment of Vcam1+ subFAPs analysis during regeneration after acute injury in

macrophage-depleted mice at the indicated time points (ITGAM–DTR mice +/- DT treatment).

Also, I was not able to find the figure legends to the supplemental figures. Therefore, I was unable to evaluate what sample sizes were used in these experiments, nor whether experiments had been replicated. Sample size issues may apply to some of this data as well, for example Suppl. Fig. 4 c,d.

Re: We are sorry to hear that. The figure legends to the supplemental figures are uploaded together combined in a unique file. Please, see the 5th Manuscript Item “Supplemental figures and legends.docx” (1135KB). As it was explained in the previous point, Suppl. Fig. 4c-d data were also reproduced in a different laboratory.

The Venn diagram figure, appearing after Fig. 1 at the end of the pdf was labeled Suppl. Figure 8. However, it appears to be out of place or mislabeled, as there are no supplemental figures 5-7.

Re: We apologize that Suppl. Fig.8 was uploaded by mistake, so it has been removed from the new resubmission.

Reviewer #3 (Remarks to the Author):

This is the revised version of Nat. Cell Biol. One of major issues raised by this reviewer was the differentiation ability of each subFAPs. In this revised

version, authors sincerely addressed to this issue. Authors also provided new data in SFig.3d (clustering of RNA-seq profiles). Based on these results, authors mentioned that each subFAPs is discriminated them from tissue fibroblasts. However, the frequency of adipogenic-induction was too low (only 20-30%). In addition, molecular signature of fibroblast might be strongly affected by their environment. In this case, cardiac fibroblast data alone might be not enough. More data derived from other tissues (non-muscle tissues) will strengthen authors' finding. Reviewer strongly recommends to include the data indicating the results of adipogenic differentiation ability of subFAPs in the main or supplemental Figure.

RE: We welcomed this reviewer's request to include additional data derived from RNAseq of fibroblasts from other non-muscle tissues (e.g. synovia, lung and skin). The inclusion of this additional data further supports our conclusion that tissue-specific fibroblasts and/or myofibroblasts from multiple sources clearly separate from FAPs, as well as from other muscle-derived cell types (SCs and macrophages), regardless of whether they derived from muscle or non-muscles tissues.

Figure legend. *Unsupervised clustering of mouse bulk FAPs (black), sub-FAPs populations (Vcam1+ in green, Tie2^{high} in blue, Tie2^{low} FAPs in red), satellite cells (grey), macrophages (purple) in the different conditions (WT, MDX, injury day 1 and day 3) and mouse tissue-resident fibroblasts (cardiac in brown, lung in cyan, synovial in yellow, dermal in magenta).*

Regarding the reviewer statement “However, the frequency of adipogenic-induction was too low (only 20-30%)” we dispute that this is not supported by a proper evaluation of our data versus previously published data. In principle, the metrics used for analysis of the data are different and therefore are not comparable. In our study, we have scored the adipogenic differentiation of subFAPS and bulk FAPs by using MetaXpress® software on the ImageXpress® Micro System (Molecular Devices), which provides an unbiased and highly reproducible analytical setting, and was not used in any of the previously published works the reviewer referred to. An illustrative representation of the figures relative to the adipogenic differentiation of FAPs published so far is shown below. Please, note that quantification of adipogenic differentiation *in vitro* using Perilipin or Oil Red O staining was not performed at all in these studies (see panels A and B from Joe et al., 2010; panel D3 from Uezumi et al., 2010; panels F and G from Woczyna et al., 2012). The only quantification available was made by counting of cells expressing the adipogenic transcription factors PPARgamma and CEBP using Win ROOF software (panel E from Uezumi et al., 2010) or by manual counting the number of clones undergoing adipogenic differentiation after clonogenic assay from single cells (panel H from Woczyna et al., 2012). Other than differences in counting methods (software-assisted vs. manual), we note that the parameters and culture conditions used to quantify FAP’s adipogenic potential in previous reports are completely different from those used in our manuscript. Expression of PPARgamma and CEBP in FAPs (panel E from Uezumi et al., 2010) refers to a pre-adipogenic stage and does not coincide with their full differentiation potential, as evaluated by Perilipin staining or Red Oil O. Percentage of clones undergoing adipogenic differentiation from single cells (panel H from Woczyna et al., 2012) entails culture conditions and quantification criteria (presence of lipid droplets, perilipin and/or Oil Red O staining) that are clearly different from those used in our paper.

With that said, in order to make the only possible comparison (i.e. comparative visual inspection of Perilipin staining) between FAPs and subFAPs from our manuscript and others previously published, we included the pictures showing Perilipin staining of our FAP cultures – see Panel I in the figure below. Top Panel I (Malecova et al., 2018 refers to unpublished data from Puri lab), shows that the frequency of

Perilipin staining in FAPs and subFAPs cultured in optimal conditions appears indistinguishable from that reported in the previous works. If the reviewer's statement refers to the lower frequency shown by most of the FAPs and subFAPs (except for Tie2^{high} subFAPs) in sub-optimal limiting conditions (low confluence cultures) that we originally showed in our first submission, then it should be acknowledged that experimental purpose of this condition, which was instrumental to reveal a higher propensity of Tie2^{high} subFAPs to adopt an adipogenic potential in sub-optimal limiting conditions (see bottom panels I, unpublished data). Nonetheless, we note that there are no reported data on FAP's adipogenesis in sub-optimal conditions (as they give low frequencies of adipogenesis in bulk FAPs), and therefore it is impossible to make any comparison here too.

Figure legend.

A) *Lin*⁻*Sca-1*⁺*CD34*⁺ cells contain adipogenic progenitors. Sorted cells spontaneously gave rise to multilocular adipocytes (center). Triglycerides were detected by oil red O staining in unilocular mature adipocytes after 30 days (right). Scale bars, 50 μ m (left and center) and 100 μ m (right). **B)** FAPs cultured in either expansion conditions or adipogenic conditions containing PPAR γ agonists for 17 days. FAPs spontaneously gave rise to perilipin positive cells in expansion conditions and, more efficiently, in adipogenic conditions. Scale bar: 100 μ m (Joe et al., 2010). **C)** Phase contrast image of CD31⁻CD45⁻

PDGFR α ⁺ cells under adipogenic culture conditions. D) Cells were stained with antibodies against PPAR γ (D1) and C/EBP α (D2), or with oil red O (D3). Insets show high magnification images. Scale bars, 50 μ m. E) Adipogenic differentiation was evaluated by quantifying the percentages of C/EBP α - and PPAR γ -positive cells at day 4 and 10 in culture. Error bars indicate mean⁺ s.d. (Uezumi et al., 2010). F) GFP⁺ CD31⁻ CD45⁻ PDGFR α ⁺ Sca-1⁺ progenitors isolated from Tie2-Cre;R26^{NG/+} adult mice muscles were sorted directly into individual wells of 96-well plates, cultured in growth medium and evaluated for adipogenic differentiation. Typical example of a live colony maintained in growth medium for 12 days. Insets are magnified views of the boxed areas. Adipocytes with multilocular lipid deposits are apparent. Hoffman modulation contrast microscopy. Scale bars represent 100 μ m (F1). 14-day culture stained with GFP (F2) and Oil Red O (F3) to definitively identify lipid droplets. (G) Typical examples of 23-day cultures maintained in growth medium and stained for Perilipin. Scale bars represent 50 μ m. (H) Quantification of adipogenic potential, which was assessed between 10 and 14 days of culture, and colonies were scored as positive if they contained cells with large lipid droplets, stained with Oil Red O, or expressed Perilipin (Wokczynna et al., 2012). I) Bulk FAPs and subFAPs (as Lin-Itga7-Sca1⁺Tie2^{low}; Tie2^{high}, or Vcam1⁺) were isolated from muscles 3 days after acute injury (3d NTX) and cultured in expansion conditions for 3 days at low confluence (lower panel) or 4 days at high confluence (upper panel) and induced to differentiate for 6 days in adipogenic conditions (3 days adipogenic induction and 3 days adipogenic maintenance). Scale bars represent 100 μ m (unpublished data, Puri Lab).

The reviewer strongly recommended “to include the data indicating the results of adipogenic differentiation ability of subFAPs in the main or supplemental figures”. However, we argue against this request, as we remain with our initial position that our manuscript goes further beyond the use of the adipogenic potential to define FAP identity, since the identification of subFAPs from single cell transcriptomic analysis provides novel criteria for a more advanced and biologically insightful definition of FAPs that would have never been made possible by relying just on the adipogenic differentiation.

In dystrophic muscle, there are many fibroblasts and myofibroblasts. In this study, authors used limb muscle of mdx mice which show less dystrophic change. Thus mdx sample might not include fibroblasts and myofibroblasts. Does subFAPs have the possibility to include fibroblasts and myofibroblasts? In order to confirm this question, diaphragm sample of mdx might provide important information.

Re: We do not fully understand the rationale of this question. In other points of his/her revision this reviewer expressed concerns about the possibility that certain sub-populations (i.e. VCAM1+ subFAPs) could be contaminated by tissue-resident fibroblasts or myofibroblasts. We therefore argue that using limb muscles is the best way to eliminate this bias, since these samples do not include fibroblasts and myofibroblasts, as he/she also pointed. By contrast, the diaphragm provides the most enriched source of potential contaminating cells. More importantly, data from diaphragm-derived FAPs would be impossible to compare to the data reported in our manuscript, which entirely refers to FAPs derived from the limb muscles. In this regard, we emphasize that our study is specifically focused on FAPs as defined in the available literature. Indeed, the large majority of the published works analyzed FAPs derived from limb muscles, while only one previous study published by Uezumi et al., 2011 has analyzed diaphragm-derived FAPs. We also note that FAPs derived from different types of muscle are unlikely comparable as the muscle type-specific differences might bias their behavior and phenotype, as observed by Formicola et al., with FAPs from extraocular muscles (EOMs) (Formicola et al., 2014).

We would like to reiterate that the major finding of our study is the identification of markers that identify distinct FAPs subpopulations that reflect dynamic transition through cellular states in response to physiological, pathological and experimental environmental perturbations. Thus, for a matter of scientific rigor we limit our analysis to FAPs from limb muscles exposed to each of these perturbations. As we provided further evidence that our subFAPs are different from tissue-specific fibroblasts and myofibroblasts (see the unsupervised clustering in Suppl. Fig. 3d) do not consider investigating further the profile/role of myofibroblasts in this study.

As reviewer pointed out, the difference between acute and chronic change of FAPs is important when considering the onset or progression of pathological changes of skeletal muscle. Authors described signatures of each subFAPs. For example, Vcam1+ and Tie2low exhibit a pro-fibrotic gene expression profile. Fibrosis-related genes are highly enriched in Tie2-low and Tie2-high, but they also express anti-fibrotic gene, BMP7. Authors explained that BMP7 might contribute to non-fibroblastic role of Tie2-low and high FAPs. However, there is no data indicating the direct evidence for the correlation between the genes and pathological change including fibrosis. Without these data, this work seems to be descriptive.

Re: We agree with the reviewer that our transcriptome analysis of the subFAPs from an acute and chronic injury environment reveals several highly interesting potential regulators of FAPs activity. We believe that future studies are necessary to address the role of these regulators in a more systematic approach. Most importantly, our study revealed that subFAPs are highly unstable when cultured *in vitro*. We argue that the rapid changes of subFAP identity observed upon culturing complicates their functional analysis *in vitro*. Therefore, future studies toward challenging functional gene networks predicted by our RNAseq will absolutely require *in vivo* approaches, such as the use of conditional cell-specific gene ablation. Such approach will entail completely new projects and are definitely not within the scope of the present manuscript. Instead, our highly reproducible RNA-seq data (two independent experiments performed with a separate cohort of animals on separate days) provide a confident source of a comprehensive information for the scientific community to allow formulation of precisely such novel hypothesis to be tested, as mentioned e.g. by the reviewer here. Nonetheless, we note that we have begun to functionally challenge the most relevant data emerged by our analysis – that is the relationship between subFAPs and signals exchanged with the inflammatory infiltrate – by using a mouse model of macrophage depletion (ITGAM mice) and by exploiting the model of inflammation-free “sterile” neonatal myogenesis.

Please see additional comments.

1: Related to FACS analyses

Authors RE: We are aware of this discrepancy. However, given that we have generated a statistically robust dataset (4 independent biological replicates in figure 2a,b,c and independent biological triplicates in figure 4d,e,f), the only explanation we can provide at this stage regards most likely the different intensity/extent of an acute injury. As we are comparing our data with only one report, we believe this contention should remain open to further experimentation from other investigators.

Authors have to stain tissue samples with anti-Pdgfra antibody, a marker for all subFAPs. Authors might observe many Pdgfra+ cells in inj-3d than these in inj-1d. The result will let authors know that the previous report is not in the case of this study or authors' FACS analyses did not perform correctly.

Re: All identified subFAPs in this study express high levels of PDGFRa (see the single cell gene expression data in the Figure 1), therefore we do not think it is necessary to repeat the analysis focusing on PDGFRa+ cells. We also provided detailed description of the controls included in FACS experiments (see Suppl. Fig 2a). The individual FACS profiles from each independent experiment are available for reviewers upon request.

We also note that our study was entirely based on the most established method of FAPs isolation, as also recently described by the authors of one of the original description of FAPs published in NCB 2010 – (Judson et al., 2017) see the abstract below.

Methods Mol Biol. 2017;1668:93-103.

Isolation, Culture, and Differentiation of Fibro/Adipogenic Progenitors (FAPs) from Skeletal Muscle.

Judson RN^{1,2}, Low M³, Eisner C³, Rossi FM³.

Abstract

Fibro/Adipogenic Progenitors (FAPs) are a multipotent progenitor population resident in skeletal muscle. During development and regeneration, FAPs provide trophic support to myogenic progenitors that is required for muscle fiber maturation and specification. FAPs also represent a major cellular source of fibrosis in degenerative disease states, highlighting them as a potential cellular target for anti-fibrotic muscle therapies. Effective and reproducible methods to isolate and culture highly purified FAP populations are therefore critical to further understand their biology.

Here, we describe a fluorescent activated cell sorting (FACS) based protocol to isolate CD31-/CD45-/Integrin- α 7-/Sca1+ FAPs from murine skeletal muscle including details of tissue collection and enzymatic muscle digestion. We also incorporate optimized methods of expanding and differentiated FAPs in vitro. Together, this protocol provides a complete workflow to study skeletal muscle derived FAPs and compliments downstream analytical, drug screening, and disease modeling applications.

2. FACS profiles of WT inj3d are different among in Figure 2a, 2d, and 4a. Are there any experimental differences among these samples?

Re: The regenerative response of muscles to the notexin-mediated acute injury of muscles varies to some extent in each experiment, which is a technical limitation intrinsic to this the experimental procedure. We have observed the same phenomenon also in our time course experiment that is represented in the figure 3e,f. In the time course experiment the error bars of the mean of three independent experiments reflect this variability. Of note, each biological experimental replicate in above mentioned figures 2a, 2d and 4a was done independently on separated days with independent cohort of animals.

3. In GO 'metabolism of DNA', Vcam1+ subFAPs shows high Z-score. However, the result of WT-inj d3 Bulk shows no score. Regardless of the fact that 60-70% 'WT-inj d3 Bulk' consist of Vcam1+ subFAPs', why did WT-inj d3 Bulk show no score?

Re: The bulk FAPs are composed of subFAPs and include also orphans (the FAPs in between the gates, excluded from the FACS analysis for accuracy). The high z-score in 'metabolism of DNA' by Vcam1+ subFAPs is probably lost because of dilution of the signal. This means that, for example, some genes might not appear as upregulated in bulk because they are downregulated in other subpopulations. Also, some other genes appear in the bulk population as dysregulated that makes that category less enriched for genes compared to the complete pool.

4. Suppl. Figure 8 and Suppl. Figure 3c is same results. Please remove the S. Fig.8.

Re: We apologize that Suppl. Fig.8 was uploaded by mistake and has been removed from the new resubmission

NEW REFERENCES

Wosczyzna, M. N., Biswas, A. A., Cogswell, C. A. & Goldhamer, D. J. Multipotent progenitors resident in the skeletal muscle interstitium exhibit robust BMP-dependent osteogenic activity and mediate heterotopic ossification. *Journal of Bone and Mineral Research* **27**, 1004–1017 (2012).

Formicola, L., Marazzi, G. & Sassoon, D.A. The extraocular muscle stem cell niche is resistant to ageing and disease. *Frontiers in Aging Neuroscience* **1**, 6-328 (2014).

Judson, R.N., Low, M., Eisner, C. & Rossi, F.M. in *Skeletal Muscle Development* **1668**, 93-103 (Springer, New York, NY, 2017).

Reviewers' Comments:

Reviewer #1:

Remarks to the Author:

I disagree with the other reviewer that 30% fat differentiation is not sufficient to demonstrate that the cells are adipogenic. It is clearly more than sufficient.

In addition, I think your other reviewer projects a model of mesenchymal cell type specification that is fundamentally unsupported in the literature. In my view, it is not reasonable to discriminate clearly between mesenchymal cells, fibroblasts, myofibroblasts and the cells being studied in this paper. These terms are used broadly, differently by different groups, and refer to highly overlapping cell types. I (and most of the field now) like the term fibroadipogenic progenitors for this reason; it seems the other reviewer does not, probably because of their own favored model of mesenchymal cell biology of muscle. Further addressing this issue risks forcing this manuscript to say things that are not really supported.

I thought this was a strong manuscript for nature cell biology, so I think it is a great publication for nature communications.

My recommendation is for acceptance without further requests for experiments. The main points of the paper are not going to be altered at this point, and the paper was substantially overhauled between first and second submission, resulting in a great improvement.

Reviewer #3:

Remarks to the Author:

Authors addressed some issues raised by Reviewer. However, unfortunately, authors did not satisfactorily address other issues.

If subFAPs retained the adipogenic differentiation ability, such data is very important for readers.

Authors stated that their analyses satisfied the criteria of the previous reports by Rossis group. On the other hand, in the first round of review, authors stated the 'We are aware of this discrepancy. However, given that we have generated a statistically robust dataset (biological triplicates), the only explanation we can provide at this stage regards most likely the different intensity/extend of an acute injury. As we are comparing our data with only one report, and we believe this contention should remain open to further experimentation from other investigators.' To my opinion, authors interpret the data by same (Rossi's) group to suit authors' purpose. Reviewer recommended authors to do the immunohistochemical studies to confirm the reliability of FACS data in this study.

In regarding the pathogenic roles of subFAPs, reviewer asked authors to use more severely damaged muscle, diaphragm. Authors used relatively less pathogenic muscle to characterize the pathogenic role of subFAPs. It means that the characteristics of subFAPs provided here DO NOT contribute to severe pathogenesis. Reviewer also wonders that what fraction includes fibroblast and myofibroblasts. It is one reason why reviewer required authors to use diaphragm muscle.

Reviewer #4:

Remarks to the Author:

Technically, this is a good study. It is innovative in the muscle regeneration field as it provides molecular evidence of subpopulations of FAPs, a very important player in the the regeneration process, as it promotes myogenesis or fibrosis depending on the physiopathological context. The identification of these subpopulations by single cell RNAseq and further utilization of newly-

identified cell surface receptors in FACS analysis is the state-of-the-art for finding new cell types. The authors have been brave and taken this approach in injured muscle, and they have succeeded. From this point of view, the paper is good. The fact that they extended the analysis on injured muscle to muscle of dystrophic mice is a plus.

The major problem in my view is that its major findings (despite being interesting and novel) are descriptive in nature. In fact, in my modest opinion, Reviewer #3 is mostly right . These new subpopulations of FAPs need to be characterized somehow further by functional assays.

I would like to indicate the following concerning this comment of Reviewer #3:

He/she says:

'In dystrophic muscle, there are many fibroblasts and myofibroblasts. In this study, authors used limb muscle of mdx mice which show less dystrophic change. Thus mdx sample might not include fibroblasts and myofibroblasts. Does subFAPs have the possibility to include fibroblasts and myofibroblasts? In order to confirm this question, diaphragm sample of mdx might provide important information.'

I think that this is an appropriate comment of the Reviewer. The authors insist that their focus is limb muscle; however, limb muscles of 3 month old mice do not represent at all fibrotic muscle tissue. Thus, analyzing these FAP subpopulations in the diaphragm will be pertinent and will improve the manuscript. The authors have the technology/mice and everything in place to do these analysis.

In summary: I don't think that these subpopulations of FAPs are sufficiently characterized. This is a must. I would not ask the authors for the distinction between fibroblasts and myofibroblasts.

Yet, I think that analyzing these subpopulations in a real fibrotic context like mdx diaphragm is needed.

Rebuttal Letter

Dear Editor

Thank you for having considered our manuscript entitled "A spectrum of cellular states within Fibro-Adipogenic Progenitors (FAPs) upon physiological and pathological perturbations of skeletal muscle" for publication in Nature Communications.

We appreciated that the manuscript was sent back to an addition referee, who had the opportunity to evaluate our reply to the other referee's comments.

This additional referee (Reviewer #4) acknowledged that "Technically, this is a good study. It is innovative in the muscle regeneration field as it provides molecular evidence of subpopulations of FAPs, a very important player in the regeneration process, as it promotes myogenesis or fibrosis depending on the physiopathological context. The identification of these subpopulations by single cell RNAseq and further utilization of newly-identified cell surface receptors in FACS analysis is the state-of-the-art for finding new cell types. The authors have been brave and taken this approach in injured muscle, and they have succeeded. From this point of view, the paper is good. The fact that they extended the analysis on injured muscle to muscle of dystrophic mice is a plus"

However, he/she also indicated that ".....analyzing these FAP subpopulations in the diaphragm will be pertinent and will improve the manuscript. The authors have the technology/mice and everything in place to do these analysis. In summary: I don't think that these subpopulations of FAPs are sufficiently characterized. This is a must. I would not ask the authors for the distinction between fibroblasts and myofibroblasts. Yet, I think that analyzing these subpopulations in a real fibrotic context like mdx diaphragm is needed".

RE: We have performed the suggested experiment, which is now described in the text, as follows

"Finally, we evaluated whether the dynamics of subFAPs typically observed in unperturbed conditions (i.e. Tie2^{low} and Tie2^{high} accounting for the vast

majority of FAPs) was skewed toward enrichment in Vcam1+ subFAPs in diaphragm of mdx mice at the very early stages of disease progression (6 weeks old mice), compared to age-matched control (WT) mice. The diaphragm is an involuntary muscle subjected to the continuous contractile activity that permits breathing. As such, it is invariably used from birth and therefore is the earliest muscle to develop fibrosis and in DMD, showing the most pronounced accumulation of fibrotic areas¹⁵. We observed that diaphragms of 6 weeks old WT mice exhibited a higher ratio between Tie2^{high} and Tie2^{low} subFAPs, compared to limb muscles used for the previous analysis, presumably because of more intense contractile activity and younger age (compare Supplementary Fig. 4f and g with Fig. 2D). Remarkably, mdx diaphragms exhibited a significant percental reduction in Tie2^{high} subFAPs and a proportional increase in Vcam1+ subFAPs (Supplementary Fig. 4f and g).”

Reviewers' Comments:

Reviewer #4:

Remarks to the Author:

The authors have satisfactorily addressed the issues that I previously raised. Thank you.